# Synthesis and Evaluation of Clinically Translatable Targeted Microbubbles Using a Microfluidic Device for In Vivo Ultrasound Molecular Imaging

**DOI:** 10.3390/ijms24109048

**Published:** 2023-05-20

**Authors:** Rakesh Bam, Arutselvan Natarajan, Farbod Tabesh, Ramasamy Paulmurugan, Jeremy J. Dahl

**Affiliations:** Canary Center for Cancer Early Detection, Department of Radiology, Stanford University School of Medicine, Palo Alto, CA 94304, USA

**Keywords:** microbubbles, bioconjugation, thiol–maleimide, phospholipids, microfluidics

## Abstract

The main aim of this study is to synthesize contrast microbubbles (MB) functionalized with engineered protein ligands using a microfluidic device to target breast cancer specific vascular B7-H3 receptor in vivo for diagnostic ultrasound imaging. We used a high-affinity affibody (ABY) selected against human/mouse B7-H3 receptor for engineering targeted MBs (TMBs). We introduced a C-terminal cysteine residue to this ABY ligand for facilitating site-specific conjugation to DSPE-PEG-2K-maleimide (M. Wt = 2.9416 kDa) phospholipid for MB formulation. We optimized the reaction conditions of bioconjugations and applied it for microfluidic based synthesis of TMBs using DSPE-PEG-ABY and DPPC liposomes (5:95 mole %). The binding affinity of TMBs to B7-H3 (MB_B7-H3_) was tested in vitro in MS1 endothelial cells expressing human B7-H3 (MS1_B7-H3_) by flow chamber assay, and by ex vivo in the mammary tumors of a transgenic mouse model (FVB/N-Tg (MMTV-PyMT)634Mul/J), expressing murine B7-H3 in the vascular endothelial cells by immunostaining analyses. We successfully optimized the conditions needed for generating TMBs using a microfluidic system. The synthesized MBs showed higher affinity to MS1 cells engineered to express higher level of hB7-H3, and in the endothelial cells of mouse tumor tissue upon injecting TMBs in a live animal. The average number (mean ± SD) of MB_B7-H3_ binding to MS1_B7-H3_ cells was estimated to be 354.4 ± 52.3 per field of view (FOV) compared to wild-type control cells (MS1_WT_; 36.2 ± 7.5/FOV). The non-targeted MBs did not show any selective binding affinity to both the cells (37.7 ± 7.8/FOV for MS1_B7-H3_ and 28.3 ± 6.7/FOV for MS1_WT_ cells). The fluorescently labeled MB_B7-H3_ upon systemic injection in vivo co-localized to tumor vessels, expressing B7-H3 receptor, as validated by ex vivo immunofluorescence analyses. We have successfully synthesized a novel MB_B7-H3_ via microfluidic device, which allows us to produce on demand TMBs for clinical applications. This clinically translatable MB_B7-H3_ showed significant binding affinity to vascular endothelial cells expressing B7-H3 both in vitro and in vivo, which shows its potential for clinical translation as a molecular ultrasound contrast agent for human applications.

## 1. Introduction

Phospholipids-based microbubbles (MBs) serve as blood pool contrast agents and are used in conjunction with contrast-enhanced ultrasound imaging for diagnostic applications in cardiovascular and cancer imaging [1]. Molecularly targeted contrast agents that bind to receptors or biomarkers expressed in vascular targets in vivo are desirable due to their ability to enhance the specificity of disease diagnosis. To facilitate this process, MBs need to be conjugated with specific antibodies or engineered small protein ligands that can direct the MBs to bind to desired molecular targets in the blood vessels. Typically, the ligands are attached to the bubbles at the liquid interface of the phospholipid-monolayer shell by chemical conjugation methods. For example, KDR-targeted phospholipid MBs are designed to target tissue-specific pathological angiogenesis and have shown promising results in clinical trials [2,3]. Currently, there are no FDA-approved targeted MBs (TMBs) available for ultrasound molecular imaging. Moreover, the FDA-approved non-targeted MBs are produced by mechanical agitation method, which produce MBs of broader size distribution that can cause heterogeneous acoustic response with variabilities in ultrasound imaging sensitivity [4].

Developing a reproducible methodology to cater the synthesis of TMBs of uniform sizes with enhanced acoustic properties can improve the imaging sensitivity while facilitating the on-demand preparation of TMBs for ultrasound molecular imaging in clinical settings. Microfluidic-based processing methods offer better control over MB size dispersity for molecular imaging applications as well as improve the production efficiency and reproducibility [5,6]. Microfluidic devices produce MBs by a variety of pinch-off mechanisms with precisely controlled flow of gas and liquid streams. Chips fabricated with specific microfluidic channel geometry can consistently maintain production uniformity of MBs compared to those generated using sonication and amalgamation techniques [5]. Microfluidics-based devices can meet the requirements for clinically translatable TMB synthesis by providing an optimal platform technology.

Lipid-shell MB formulations consist of PEGylated phospholipids with functional groups (biotin/streptavidin, NHS esters, maleimide) that allow for the conjugation of antibodies or proteins to the MB shell. Surface-functionalization of proteins is often performed in pre-formed MBs with highly efficient chemical conjugation methods (streptavidin-biotin [7], NHS ester-amine [8], thiol–maleimide [9,10,11,12]); however, the MB labeling is not fully controlled in this approach, resulting in batch-to-batch variations. Further, ligand labeling in pre-formed MBs can affect the stability of MBs, and it is not a feasible strategy for clinical practice because technologists need readily injectable molecular contrast agents for patient imaging. Random chemical conjugations between phospholipids and antibodies, or proteins with multiple reactive side chains (e.g., lysine amino acids) can reduce the overall biological activity of the conjugated ligands [13]. To produce uniformly labeled TMBs for clinical use, phospholipid–ligand bioconjugates can be prepared prior to their integration into MBs in a controlled manner. Stable bioconjugate preparation using clinically applicable site-specific covalent bonding strategies between proteins and lipids can further enhance uniform MB labeling with targeting ligands [14]. MBs formulated with the incorporation of lipo-peptide bioconjugates binding to the KDR receptors have previously been applied with covalent amine conjugation chemistry [15]. Unlike larger proteins such as antibodies, engineered small protein ligands can be easily modified to promote site-specific bioconjugation reactions that are scalable and safer for therapeutic and imaging applications [16,17]. For example, small proteins can be modified to display a terminal cysteine residue, which upon reduction forms a stoichiometrically controlled and stable thioether bond to molecules with maleimide functional groups [18]. The site-specific covalent bonding methods could preserve ligand functionality while reducing variabilities in bioconjugation reactions, enhancing targeted MBs performance, and reproducibility.

Recent developments in microfluidic-based processing methods offer better control over the size distribution of the MBs. In addition, this tool has also shown reproducible production efficiency and size distribution [5,6]. Microfluidic devices produce MBs by a pinch-off mechanism through tapered flow channels with precisely controlled flow of gas and liquid streams. The microfluidic device used here can consistently maintain uniform production of desired MBs traits compared to those generated using sonication/amalgamation techniques [5]. Peyman et al. reported that MBs produced by a flow-focusing microfluidic device have higher ultrasound scattering properties over a wide range of frequencies compared to those formed by mechanical agitation methods [19]. In addition, microfluidic techniques can be applied to produce suspensions of uniformly sized MBs with contrast properties [20] tailored to specific ultrasound frequencies for improved imaging [21]. For contrast-enhanced molecular imaging applications, labeling consistency of ligands on the surface of the MB shell is important for the optimal target-binding performance of targeted MBs. Furthermore, ligand labeling distribution can be enhanced upon preferential use of phospholipid formulations and specific lipid handling methods prior to MB production [22,23]. Microfluidic devices allow precise control over these production parameters to achieve standardized methodologies of targeted MB synthesis.

Herein, we present a clinically translatable synthesis of TMBs, with an optimized workflow using a microfluidic system to directly incorporate targeting ligand or bioconjugates. In this TMB, the targeting moiety is an engineered affibody (ABY) protein selective against an angiogenic biomarker, the B7-H3 (CD276). B7-H3 is a type I transmembrane protein ligand with 316 amino acids (45–66 kDa) and belongs to a family of immune checkpoint molecules [24]. Several studies have shown that B7-H3 plays a role either as a co-stimulatory or as a co-inhibitory in T cell-mediated adaptive immunity [25]. However, when compared to tumor tissue, the level of expression is much lower in most other normal organs and tissues. In addition to tumors, it is predominantly expressed on the surface of T and B cells [24,26]. It has been reported that several cancer types express B7-H3 at higher level [20,27] which include prostate cancer [28,29], renal cell carcinoma [30], ovarian cancer [31], glioblastoma [32], osteosarcoma [33], pancreatic cancer [34], neuroblastoma [35], diffuse intrinsic pontine glioma, and mesothelioma [36,37].

The ABY is engineered to express a cysteine residue at the C-terminus for a site-specific conjugation with phospholipids (DSPE-PEG-maleimide; 2 kDa) with an optimized thiol–maleimide chemical reaction to form stable bioconjugates and to be stable at physiological temperatures and pH. This novel TMB was characterized in terms of their size distribution, concentration, ligand display, and target-binding performance in vitro and in vivo. Our workflow consistently results in uniform MB size, yield, and binding activity to its cellular target, B7-H3, expressed by vascular endothelial cells (Figure 1).

## 2. Results and Discussion

### 2.1. Expression and Target Binding Assessments of ABY Ligands after Site-Selective Protein Modifications

After successful engineering, we tested the molecular weight of both the binders by MALDI-TOF (Figure 2A) and binding affinities to human/mouse B7-H3. Binding affinities for ABY_AC2_ and ABY_AC12,_ which showed K_d_ of 310 ± 100 nM and 0.9 ± 0.6 nM, respectively. Similarly, purities of both ABY_AC2_ and ABY_AC12_ protein were assessed by the SDS-PAGE (Figure 2B). The result shows one major band corresponding to ~9 kDa for AC12 (lane #3) compared to AC2 (lane #2), which shows an expected band at ~9 kDa and an additional high molecular weight band that could be possibly due to some aggregates. Further, we tested the binding efficiency of ABY_AC2_ and ABY_AC12_ to B7-H3 in MS1 mouse endothelial cells engineered to overexpress human B7-H3 (MS1_hB7-H3_) by flow cytometry analysis. The result shows greater binding affinity by both engineered ABYs to endothelial cell line expressing B7-H3 compared to control MS1_ctl_ cells (Figure 2C). ABY_AC12_ showed low non-specific binding to MS1_WT_ cells compared to ABY_AC2_, suggesting its higher binding specificity to B7-H3.

Preclinical characterization of ligands is important before their selections for clinical studies. Hence, we tested the binding efficiency of ABY ligand to MDA-MB-231 human breast cancer cells expressing 4Ig B7-H3 isoform, and 4T1 murine mammary cancer cells expressing 2Ig B7-H3 isoform. We confirmed both of these cell lines for their corresponding B7-H3 isoforms expression by anti-B7-H3 antibody staining based flow cytometry assay (Figure 2D). Interestingly, the binding efficiency of both ABY_AC2_ and ABY_AC12_ to 4T1 cells expressing murine B7-H3 was similar; however, the binding efficiency of ABY_AC12_ to human B7-H3 expressed on MDA-MB-231 cells was significantly higher compared to ABY_AC2_. These results are consistent with our previous findings that the engineering of ABY_AC12_ variant by affinity maturation of ABY_AC2_ against human B7-H3 resulted with enhanced target specificity, thermal stability, and refolding ability of the engineered ABY_AC12_ ligand [38].

Bacterial tags, such as His-Tag, should be removed from purified proteins being considered for clinical applications as their administration in humans can trigger immune-related adverse events. We tested the ABY_AC12_ for His-Tag removal in a proof-of-concept enterokinase enzyme-mediated cleavage assay to produce a clinically applicable protein ligand. Incubation of ABY in the cleavage buffer overnight caused dimer formation with a small amount of non-specific aggregation as observed in the SDS-PAGE analysis (Appendix A); however, the enterokinase enzyme was able to remove the His-Tag from both the dimer and monomer forms of the modified ABY consisting of enterokinase recognition sequence (note the relative reduction in ABY molecular weight after cleavage), but it did not remove the His-Tag from the ABY protein that does not have the recognition sequence. It is possible to minimize the ABY dimer formation by thiol reduction at its cysteine residue prior to incubation with enterokinase, and by decreasing the reaction incubation period (Figure 2B, lane #3). Based on all these results, we selected the ABY_AC12_ as a clinically translatable ligand for further downstream formulations and applications.

### 2.2. Bioconjugation Optimization and Stability of the ABY-Phospholipid Conjugates

The site-specific conjugation of ABYs with Mal–phospholipids via thiol–maleimide addition reaction yielded >90% conjugation efficiency. Modification with cys with more than one cys moiety in a molecule provides multiple conjugation reaction leads to several derivatives [39]. Regarding our ABY binders (ABY_AC2_, and ABY_AC12_), there is no cys residue in both except the one introduced by us at the C-terminus to control the conjugation reactions with only one conjugate derivative. After this modification, we optimized conjugation condition by using His-Tagged ABY_AC12_ to test its ability to conjugate with the Mal functional group at neutral pH (7.0). First, the c-terminal cys in the ABY was reduced to activate the thiol group for its conjugation with Alexa Fluor 647-Mal dye. A significant fraction of reduced ABY_AC12_ was able to conjugate with the Mal-bearing dye at 10-fold molar excess of the dye as evidenced by fluorescence imaging of the band after resolving in SDS-PAGE gel, whereas no fluorescence band was visualized with the ABY sample devoid of the conjugating dye (Figure 3A, lane #3). ABY binder with DSPE-PEG-Mal at 1:10 molar ratio yielded a high conjugation efficiency after 2 h of reaction at room temperature (Figure 3B, lane #4). A homogeneously stained diffused band in Figure 3B represents the phospholipid of expected molecular weight of 2.9 kDa.

SDS-PAGE analysis in Figure 3B displays bands corresponding to the conjugates sampled from various molar ratio (1:10, 1:15, 1:20) reactions and compared them with lipid–Mal without any conjugated (Figure 3B, lane #2) and unconjugated ABY (Figure 3B, lane #3). While lipid–Mal displays a single band, the reduced ABY shows >98% monomer form with a small fraction of ABY-cys forming dimers during the incubation period. Regarding ABY-reduction, at 1:20 molar ratio, no ABY was leftover as evidenced by no band for monomer, whereas at 1:10, and 1:15 molar ratios, ABY monomer appearance showed proportionally decreased bands corresponding to 17 kDa (Figure 3B), and the lipid–Mal monomer showed increased band appearance corresponding to 11 kDa (Figure 3B). These results demonstrate that 20-fold molar excess of lipid–Mal yields 100% conjugation efficiency to form ABY-DSPE-PEG. In addition, MALDI-TOF analysis of ABY-conjugate accounted for an approximate peak size at 11.307 kDa, which is closer to the theoretical summation of molecular weights of DSPE-PEG-Mal lipid monomer (~2.9 kDa) and ABY (~8.3 kDa) (Appendix A). At pH7, the efficiency of the thiol–maleimide reaction was higher; however, further increases in pH could reduce efficiency by yielding random amine-maleimide reactions [40]. This competing reaction may lead to the formation of undesirable DSPE-PEG-ABY conjugate, which in turn can decrease the chemoselectivity of Mal to the terminal cys on ABY. To confirm the cys site-specificity of the thiol–Mal adducts at neutral pH, we repeated the conjugation of DSPE-PEG-Mal with thiol reduced or non-reduced ABY in an overnight (~16 h) reaction at 4 °C. As expected, a strong band of bioconjugation was observed in sample containing the DSPE-PEG-Mal and reduced ABY monomer, but the longer reaction time also increased the amount of ABY dimers and high molecular weight conjugates (~30 kDa) (Figure 3C). Although a minor bioconjugate band (~11 kDa) was also observed between DSPE-PEG-Mal and non-reduced ABY monomer, a major conjugate band (~25 kDa size) appeared above the ABY dimer band, which suggests the predominance of amine–Mal adduct formation when reactive thiols from cysteine are not available. The relative increase in the amount of ABY dimer under non-reducing conditions may have contributed to increased Mal reaction with lysines from ABY dimers. Furthermore, the higher molecular weight conjugates (~30 kDa) also appeared with non-reduced ABY, suggesting that the conjugations under prolonged reaction times lead to Mal-amine reaction. The reduced ABY protein did not form bioconjugates in the reaction samples consisting of control DSPE-PEG without Mal functional groups.

We further tested the stability of the DSPE-PEG-ABY under several storage conditions. Conjugate samples were concentrated and dissolved in water for analysis (Appendix A). These samples were stored in two different conditions, at 4 °C or as a lyophilized powder at −20 °C. Bulk storage of lyophilized conjugate powder allows DSPE-PEG nanomicelles to be preserved without loss in integrity [41]. Overall, stability testing of DSPE-PEG-ABY indicated that the lyophilized powder conjugate showed no sign of degradation as observed in the SDS-PAGE (Appendix A). In addition, the ABY_AC12_ maintains its target binding activity and does not degrade after multiple freeze–thaw cycles. Similarly, storage of the bioconjugate in water at 4 °C for 2 weeks did not cause any significant dissociation of ABY from DSPE-PEG-ABY due to suspected retro-Michael reaction [42] or spontaneous protein degradation; however, increased non-specific aggregation was observed in this stored micellar solution (Appendix A).

Bioconjugation stability is also critical for in vivo applications where phospholipid–Mal exchange from conjugated proteins may occur due to the availability of reactive thiols or abundant amine pools in human plasma (albumin and low molecular weight reactive species) under variable physiological conditions [43]. Although such exchange reactions are shown to negatively affect the long-term therapeutic efficacy of circulating antibody–drug conjugates [43], these reactions may not be rapid enough to affect the diagnostic molecular imaging applications that rely on MBs as contrast agents with short elimination half-life of a few minutes from human circulation [2]. To further assess the in vitro stability of DSPE-PEG-ABY, we suspended this conjugate in a buffer with SDS and reactive amines and stored for 3 weeks at 4 °C. We observed only a small fraction of ABY dissociated from the DSPE-PEG-ABY under this condition, as measured by SDS-PAGE (Appendix A), suggesting that the destabilization of bioconjugate depends on storage methods and duration. Our optimized thiol–maleimide reaction conditions yielded a stable ligand–phospholipid bioconjugates that can be immediately used or preserved for long-term storage by freeze-dry for their judicious use in MB production.

### 2.3. Preparation of Targeted MB_B7-H3_ with DSPE-PEG-ABY

We prepared B7-H3-targeted MBs (MB_B7-H3_) by two different methods using the same formulation which include vialmix based amalgamation method, and microfluidic device-based bubble generation, and measured the particles size distribution for comparison (Appendix A). The microfluidic device capable of producing gas microspheres with precise control over liquid [DSPE-PEG and DPPC liposomal mixture at 1:99 or 5:95 mole % ratio of the lipid components] and gas (perfluorobutane) flow rates [5,44] (Figure 4A). A 5–9 molar % limit of DSPE-PEG coating is desirable for stable MB generation [45,46], which acts both as an emulsifier and a targeting component. MBs instability may occur due to excessive phase separation of lipids (demixing of DPPC and DSPE-PEG) on MB shell) and result in heterogeneous ligand distribution at higher percentages of DSPE-PEG [47]. DSPE-PEG-biotin emulsifier resulted in a homogeneous distribution of co-localizing ligands on MB surface when DPPC was used as the main phospholipid [23]. As the nature of ligand distribution on phospholipid shell can also affect the functionality of targeted MB and acoustic behavior, we limited the final emulsifying lipid ratio to 5 mole % during MB production. The indicated 1–5 mole % of emulsifier lipid (DSPE-PEG-Mal) micelles used in our targeted MB formulations include the total of DSPE-PEG-Mal bound to ABY_AC12_ as well as its excess in the bioconjugation reaction mixture (Figure 3B). The unconjugated lipid consisting of Mal moieties is not active due to the Mal group’s susceptibility to rapid hydrolysis and inactivation to corresponding succinamic acid in aqueous solutions [48,49]. Due to the flow-focusing microfluidic chip configuration and a pressure drop at the chip nozzle [5], we anticipated that flow of the phospholipid suspension containing a mixture of bioconjugate micelles with DPPC liposomes will spontaneously graft associated ligands into the MB shell during the pinch-off stage of MB production by microfluidic system. Particle size quantification showed no significant differences in concentration (~1.2 × 10^9^ MBs/mL) and size distribution (mean diameter = 1.3 µm; range 0.5–5 µm) between MB_B7-H3_ and control MB_NT_ (Figure 4A).

No undesirable particle aggregation was observed due to the inclusion of ABY targeting binder when the MB_B7-H3_ were examined under microscope. For the targeted MB_B7-H3,_ the average number of antibody-based ligands attached per MB is reported to be between 1.0 × 10^5^ [50] and 2.4 × 10^5^ [51], whereas small-sized protein ligands can be attached at greater numbers for enhanced binding avidity [52]. Assuming that all reduced ABY molecules are available in a monomer form during our phospholipid bioconjugation steps at an absolute reaction efficiency, a uniformly labeled monodispersed population (based on indicated particle mean size and concentrations) of MB_B7-H3_ should display approximately to a total of 4.0 × 10^9^ ABY molecules/MB with the use of 5 mole % ratio of bioconjugate mixture. To confirm ligand display on the targeted MBs, we performed flow cytometry analysis using ABY ligands with His-Tag (Figure 4B). The APC dye conjugated anti-His-Tag antibody was used as an ABY detection antibody. The flow cytometry signal was significantly higher in the MB_B7-H3_ compared to the control MB_NT_ (no binder attached) when 5-mole % of emulsifying lipids were used, suggesting that the bioconjugate integration was successful during microfluidic production. This was further supported by fluorescence microscopy of lipophilic dye-prelabeled MBs showing ABY-associated signal on the surface of the MB_B7-H3_, but not the MB_NT_ (Figure 4C). A 1-mole % bioconjugate in our experiments was not sufficient in detecting ligand display on the targeted MB shell as measured by flow cytometry. The control MBs did not show anti-His-Tag signal enhancement when free unconjugated ABY was supplied in the liquid phase consisting of DSPE-PEG control lipid and DPPC, suggesting that the ABY proteins are covalently tethered to the surface of the targeted MBs through PEG-Mal spacers in bioconjugates, and unconjugated ABY dimers. In contrast, when it is present in low amounts, it does not self-integrate into MB shell during production. To test the robustness of our lipid formulation technique, we also validated the production of targeted MB_B7-H3_ by the traditional mechanical agitation method (Vialmix, Lantheus, N. Billerica, MA, USA) using the exact lipid formulation applied to the microfluidic device. As expected, the His-Tag signal corresponding to surface-displayed ABY was also enhanced in these targeted MB_B7-H3_ compared to the control MB_NT_ in the flow cytometry assay (Figure 4D). These results indicate that molecularly targeted MB_B7-H3_ can be produced in a consistent manner by microfluidic devices using phospholipid and bioconjugate formulations.

### 2.4. Binding Validation of MB_B7-H3_ Targeted to Endothelial Cells In Vitro

We further tested the retention of target-binding properties of ABY ligand conjugated directly to phospholipid micelles or in the MB_B7-H3_ shell in vitro in cell culture settings. The target binding of DSPE-PEG-ABY bioconjugate and DSPE-PEG control micelles were tested by flow cytometry in MS1_WT_ and MS1_B7-H3_ endothelial cells. The bioconjugate binding signal (His-Tag) to MS1_B7-H3_ cells was higher than its background binding to the wild-type control cells (MS1_WT_), whereas the binding of the control micelles to both MS1 cell types (MS1_B7-H3_ and MS1_WT_) was comparable to the background binding of the bioconjugate micelles (Figure 5A). These results indicate that the conjugation of ABY_AC12_ to lipid micelles does not alter its high binding specificity to cellular B7-H3.

The cell-binding ability of the MB_B7-H3_ was assessed using a monolayer of MS1 cells and quantified the cell-bound MBs by microscopy after the removal of unbound MBs. The average number of MB_B7-H3_ binding to MS1_B7-H3_ cells per field of view was significantly higher (354.4 ± 52.3; *p* < 0.05) compared to their binding to the control MS1_WT_ cells (36.2 ± 7.5) (Figure 5B,C). In contrast, the control MB_NT_ binding to both cell types were comparable and significantly (*p* < 0.05) lower than the binding of MB_B7-H3_ to MS1_B7-H3_ cells (37.7 ± 7.8 for MS1_B7-H3_ and 28.3 ± 6.7 for MS1_WT_ cells), indicating a target-specific binding performance of MB_B7-H3_ in vitro.

### 2.5. Binding Validation of MB_B7-H3_ Targeted to Tumor Vascular Endothelial Cells In Vivo

Intravascular target-binding validation of fluorescently labeled MBs (green) was performed in a transgenic (FVB/N-Tg(MMTV-PyMT)634Mul/J) mouse with mammary tumors by bolus injection of MB_B7-H3_ via tail vein. Ex vivo analysis of tumor tissues after MB_B7-H3_ administration showed higher level of vascular B7-H3 (red; Figure 5D) in the mammary tumors as measured by immunofluorescence staining [53]. B7-H3 signal was not detected in surrounding non-tumor tissues (Appendix A). MB_B7-H3_ signal overlapped with anti-B7-H3 antibody-based immunostaining results as imaged by confocal microscopy, indicating that the MB_B7-H3_ is localized within the B7-H3-positive tumor vasculature (Figure 5D). Under the influence of hemodynamic forces in vivo, MB_B7-H3_ target binding activity is expected to be heterogeneous as observed in microscopic images of tumor tissue areas with B7-H3-positive vessels but no MB signal toward the core of larger tumors with few blood vessels. This result suggests that target binding kinetics for MBs could be more pronounced toward the tumor–host interface with abundant blood vessels and associated contrast agent signal for molecular imaging. This, in turn, may depend on the tumor perfusion heterogeneity as well as other characteristics such as tumor size and type [54]. In contrast, MB_B7-H3_ signal was not present in other highly perfused normal organs such as kidneys, which lack B7-H3 expression in their blood vessels (Appendix A). While the vascular B7-H3 signal was also absent in the liver, we observed a diffused MB signal in microscopy imaging. Liver is a highly vascularized organ and is known to rapidly remove nanobubbles/MBs from the circulation through sinusoids and the tissue-resident phagocytes [55]. This is consistent with prior observations showing the distribution properties of MBs and their clearance primarily by the liver [56]. Together, our results indicate that the targeted MBs produced by microfluidics can bind specifically to their endothelial target-receptor in vitro and tumor-associated vasculature in vivo.

Further studies utilizing these bioconjugates to formulate targeted microbubbles with different size dispersity [57] and shell properties can improve their contrast imaging performance and in vivo stability [58]. The use of engineered protein scaffolds with terminally expressed cysteine (affibodies, single-chain variable fragments, diabodies, etc.) to form phospholipid bioconjugates with variations of thiol conjugation chemistry [59,60] may further improve process control and batch reproducibility compared to post-labeling techniques for targeted MB production.

## 3. Materials and Methods

### 3.1. Reagents and Chemicals

All lipids used for microbubble (MB) formulation were purchased in powdered forms from Avanti Polar Lipids and stored at –20 °C (Avanti Polar Lipids Inc., Alabaster, AL, USA). The lipid 1,2-distearoyl-sn-glycero-3-phosphoethanolamine-N-[methoxy(polyethylene glycol-2 kDa)] (DSPE-PEG) without or with maleimide group (DSPE-PEG-Mal), along with 1,2-dipalmitoyl-sn-glycero-3-phosphocholine (DPPC) were used for our synthesis process. We also used the following other chemicals of analytical grades for MB production: Glycerol (Sigma Aldrich, Burlington, MA, USA), Sodium Chloride (NaCl; Fisher Bioreagents), Perfluorobutane (FluoroMed Inc., Round Rock, TX, USA), Perfluorohexane (Sigma Aldrich, Burlington, MA, USA), and UltraPure Distilled Water (Invitrogen, Waltham, MA, USA).

### 3.2. Cell Culture

MILE SVEN 1 mouse vascular endothelial cells of wild-type (MS1_WT_) were obtained from American Type Culture Collection (ATCC; CRL2279) and stably transfected with human B7-H3 expression vector to generate MS1_B7-H3_ cells as described previously [61]. DMEM (Corning Inc., Glendale, AZ, USA) cell culture media containing 5% fetal bovine serum and 100 units/mL of penicillin and 100 µg/mL of streptomycin was used to culture MS1 cells. Similarly, 4T1 (ATCC; CRL2539) mouse breast cancer cells and MDA-MB-231 (ATCC; HTB-26) human breast cancer cells were maintained in DMEM supplemented with 10% fetal bovine serum, 100 units/mL of penicillin, and 100 µg/mL of streptomycin.

### 3.3. Protein Modification, Expression, and Purification

ABY proteins bind to the extracellular domain of human B7-H3 receptor were previously identified using a yeast display library [61]. Parental AC2 (K_d_ = 310 ± 100 nM) binder was modified by site-directed mutagenesis to identify an AC12 binder with single digit nanomolar affinity (K_d_ = 0.9 ± 0.6 nM) [61]. We introduced a C-terminal cysteine (cys) residue tethered to a pentaglycine bridge (Gly-Gly-Gly-Gly-Gly-Cys) for site-specific bioconjugation, and an N-terminal His-Tag sequence for affinity chromatography isolation with a linking enterokinase cleavage site for His-Tag removal (Figure 2a). The purified PCR amplicons were subcloned into pET-22b plasmid vector at NheI and BamHI restriction cloning sites for protein expression and purification in *E. coli*. The sequence confirmed clone using a T7 terminator primer (Sequetech, CA, USA) was used for protein expression.

The two ABY binders (AC2 and AC12) transformed into BL21 *E. coli* (NEB) cells were used for protein purification. The bacterial colonies were outgrown in lysogeny broth supplemented with Ampicillin antibiotics for overnight, and induced for expression by treating with 200 µM of isopropyl β-D-1-thiogalactopyranoside for four to five hours at 37 °C. Bacterial cells were pelleted and lysed in lysis buffer containing 3.4 mM of NaH_2_PO_4_, 46 mM of Na_2_HPO_4_, 25 mM of imidazole, protease inhibitors (all reagents were purchased from Fisher Bioreagents, Hampton, NH, USA), 0.5 M of NaCl, 0.7 M of glycerol, and 5 of mM CHAPS (Sigma Aldrich, Burlington, MA, USA). ABY was purified by Ni-NTA affinity chromatography column (HisTrap-1 mL; GE Healthcare). Purified ABYs were desalted using 7 kDa molecular weight cut-off Zeba spin columns (Thermo Scientific, Waltham, MA, USA). ABYs were lyophilized overnight using a vacuum freeze dryer (Labconco) and stored at –20 °C until further use. Ligands were validated for purity and size by 4–12% SDS-PAGE (Novex^TM^, Thermo Scientific, Waltham, MA, USA) by loading 5 µg of protein in Laemelli buffer (Bio-Rad Laboratories, Hercules, CA, USA). The gels were stained with Coomassie G-250 dye (SimplyBlue SafeStain, Invitrogen, Waltham, MA, USA) for 1 h, followed by overnight destaining in water for visualization of protein for purity. Protein bands were visualized by gel imaging at 700 nm wavelength in an Odyssey Imaging System (LI-COR Bioscience, Lincoln, NE, USA). ABY was further characterized by matrix assisted laser desorption ionization-time of flight mass spectrometry (MALDI-TOF) to confirm sample purity and mass.

### 3.4. Biotinylation of ABY

ABY proteins were biotinylated with NHS-PEG_4_-Biotin (ThermoFisher Scientific, Waltham, MA, USA) using 1:1.6 molar ratio [61] of ABY: NHS-PEG_4_-Biotin. Briefly, NHS-PEG_4_-Biotin was dissolved in PBS (pH 7.2) and immediately mixed with 25 μL of ABY (1 mg/mL in UltraPure Water) to achieve 1.6-fold molar excess of biotin to the ABY protein and incubated at room temperature for 30 min. Unconjugated excess biotin was cleaned by passing through a 7 kDa molecular weight cut-off Zeba spin column.

### 3.5. His-Tag Removal by Enzymatic Cleavage

The N-terminal poly histidine tag (6xHistidine; His-Tag) was cleaved off from the ABY protein by recombinant bovine enterokinase enzyme (GenScript, Piscataway, NJ, USA) at the cleavage site (-Asp-Asp-Asp-Asp-Lys sequence) designed at the upstream of the ABY sequence. Briefly, freeze-dried ABY was resuspended in UltraPure Water at 1 µg/μL concentration. A total of 10 µg of ABY was added to the cleavage buffer containing enterokinase enzyme (0.4 IU or 4 IU) and incubated at room temperature for 16 h [62]. A total of 40 kDa of control protein supplied with the kit was used as a positive control while ABY expressed from the original plasmid without the N-terminal His-Tag, and cleavage site was used as negative control for the cleavage assay.

### 3.6. Phospholipid–Ligand Bioconjugation (Figure 1)

ABY was conjugated to Alexa Fluor 647-maleimide (Mal) dye (ThermoFisher Scientific, Waltham, MA, USA) in UltraPure Water at neutral pH. Prior to Mal dye conjugation, ABY was treated with TCEP.HCl (Thermo Scientific, Waltham, MA, USA) at 1:10 mole ratio for 30 min in UltraPure Water at room temperature to reduce any possible ABY dimer with Cys–Cys bond. Ten-fold molar excess of Alexa Fluor 647-Mal was added to the reduced ABY binder and incubated for 2 h at room temperature. The ABY-Alexa Fluor 647 conjugation was confirmed by fluorescence imaging of SDS-PAGE gels in the Odyssey Imaging System (LI-COR Biosciences, Lincoln, NE, USA).

DSPE-PEG-Mal was conjugated to reduced ABY in UltraPure Water at neutral pH to generate the DSPE-PEG-ABY. Briefly, ABY was first reduced as described earlier and mixed with DSPE-PEG-Mal micelles that were prepared in UltraPure Water above the critical micellar concentration (0.36 mg in 500 µL water). Dispersions of DSPE-PEG in pure water have low aggregation number and spherical core shape [63]. Conjugation reaction was optimized by performing various molar ratios of DSPE-PEG-Mal: ABY (20:1, 15:1, 10:1), respectively, for 2 h at room temperature. The effect of incubation time and temperature were also tested by performing these bioconjugation reactions overnight at 4 °C with a 20-fold molar excess of phospholipids. The resulting conjugate was purified from TCEP and other impurities using an ultracentrifugal filter column (Amicon, 10 kDa cut-off, Burlington, MA, USA). The concentrated bioconjugate in water was stored at 4 °C for the short-term, or at −20 °C as a lyophilized powder prepared by vacuum freeze–drying process for long-term.

Formation of thiol adducts can be detected by gel shift assays [48]. We also evaluated the bioconjugate reaction between ABY and the DSPE-PEG-Mal by SDS-PAGE analyses by comparing the samples with individual reaction components (ABY or DSPE-PEG-Mal) against the final product of DSPE-PEG-ABY. Approximately 5 µg of ABY and 20 µg of micellar phospholipid equivalents were loaded in Laemelli buffer for SDS-PAGE analysis (NuPAGE Bis-Tris gels with neutral pH) to compare the bioconjugates. Protein and micellar phospholipid bands were visualized by Coomassie G-250 dye staining and imaging as described above.

For the bioconjugate stability tests, DSPE-PEG-ABY bioconjugate was stored in a Laemelli buffer with SDS and competing primary amines (15 mM Tris.HCl solution at neutral pH) for 3 weeks at 4 °C, and ABY dissociation from the bioconjugate was assessed by SDS-PAGE analysis. The bioconjugates stored in water at 4 °C or as lyophilized powder at −20 °C (after resuspension in water) were also examined for degradation by the SDS-PAGE analysis.

### 3.7. Flow Cytometry Analysis

ABY binder was tested for its binding ability to MS1_WT_, MS1_B7-H3_, MDA-MB-231, and 4T1 cells. Approximately 1 × 10^6^ cells of each cell type were incubated with or without biotinylated ABY (0 and 1 µM) for one hour at room temperature, then washed 3 times in PBS containing 0.1% BSA, and incubated with streptavidin-Alexa Fluor 647 dye (Thermo Scientific) for 30 min. A control group of cells incubated only with streptavidin-Alexa Fluor 647 dye (without ABY) was used as a control for non-specific dye binding to cell surface. Cells were also tested with anti-B7-H3 antibody with APC dye (BioLegend) known as a positive control to bind for B7-H3. Next, cells were washed 3 times in 1% PBSA to remove unbound dye. A Guava easyCyte flow cytometer was used to analyze these cells using FlowJo 2.0 software for the histogram comparison of cell bound ABY signal.

In addition to ABY-PEG-dye, ABY-conjugate micelles were also tested for its target binding on the MS1 cells. We incubated 25 µL of ABY-DSPE-PEG with 0.5 × 10^6^ MS1_WT_ or MS1_B7-H3_ cells in 100 µL of PBS (corresponds to 4 µM ABY) for 1 h at room temperature. Cells incubated with DSPE-PEG micelles alone were used as a negative control. The cells were washed 3 times in 1% PBS to remove unbound micelles and incubated with anti-His-Tag-APC secondary antibody (BioLegend, San Diego, CA, USA) for 30 min. The cells were then washed 3 times in 1% PBS and analyzed using flow cytometry.

### 3.8. Preparation of Targeted Microbubbles by a Microfluidic System

The Horizon Microbubble Maker system (purchased from the University of Leeds, U.K.) was used to generate MBs. The Horizon system is a microfluidics-based system for producing uniform and reproducible distributions of polydisperse and monodisperse microbubbles by mixing lipids and gases via (interchangeable) microfluidic cartridges. This computer-controlled system can produce bubble of different sizes by adjusting the flow rate of lipid mixture, gas pressure, and perfluorobutane level (see Figure 2). Similarly, the system can also be operated using cartridges of various designs to prepare MBs of different sizes and properties. A detailed design and operation of the Horizon system is reported by Abou-Saleh et al. [5,6].

A standard operation procedure (SOP) established by the manufacturer was followed to prepare MBs [5]. We prepared DPPC based unilamellar liposomes from a thin film of phospholipids prepared in a glass vial by evaporating organic solvents from the lipid mixture by passing steady flow of N_2_ gas over this mixture. This lipid film was redissolved in saline by probe tip sonication (Branson SLPe Digital Sonifier) on ice. After sonication the liposome size was determined by Dynamic Light Scattering (DLS, Malvern Zetasizer, Malvern analytical) and Nanoparticle tracking analysis (Nanosight, NS300, Malvern). The B7-H3-targeted MBs (MB_B7-H3_) were prepared from the mixture of DSPE-PEG-ABY and DPPC (2 mg/mL of lipids containing DSPE-PEG-ABY and DPPC at 1:99 or 5:95% molar ratio) in final buffer containing 4 mg/mL of NaCl, 1% perfluorohexane and 1% glycerol (Sigma Aldrich, Burlington, MA, USA) in UltraPure Water. Briefly, DPPC thin film was hydrated in 0.5 mL of buffer in a glass vial followed by 30 min heating at 55° Celsius for lipid phase transition, 1 min vortexing, and tip-sonication on ice until the lipid solution turned transparent by visual inspection. A total of 0.5 mL of the DSPE-PEG-ABY bioconjugate solution was added to 0.5 mL of DPPC liposome buffer solution and mixed. A total of 10 µL of perfluorohexane was added to the resulting 1 mL of phospholipid mixture and vortexed immediately before MB production.

The microfluidic chips in the Horizon system were supplied with perfluorobutane gas (FluoroMed, Inc, Round Rock, TX, USA) at 1000 mbar pressure through a gas inlet channel and the 1 mL of lipid mixture was introduced in aqueous phase through two opposing inlet channels at 100 µL/minute flow rate, with the flow focused through a chip nozzle to produce MBs. Control MBs (Non-targeted = MB_NT_) were prepared with the lipid mixture containing DSPE-PEG without Mal group and DPPC at the indicated mole ratios described above. MB count and particle size were determined by Accusizer 770A (Particle Sizing Systems) and the values were compared to MB_NT_.

### 3.9. Validation of ABY Display on Targeted Microbubbles

The DSPE-PEG-ABY conjugate amalgamated into the DPPC MBs was confirmed by flow cytometry with an anti-His-Tag-APC antibody. We added 5 µL of antibody to 100 µL of MBs (MB_B7-H3_ or MB_NT_) containing 1 × 10^8^ particles and the mixture was incubated at room temperature for 1 h. MBs were washed 3× in 500 µL of PBS using a microcentrifuge at 300 *g* for 3 min. After each wash, the upper milky layer of floating MBs was carefully separated by removing the liquid wash with a syringe needle and resuspending MBs in fresh PBS. ABY (ABY_NoHis-Tag_) displayed on the surface of MB_B7-H3_ was detected by flow cytometry (Guava easyCyte) of MB-bound anti-His-Tag-APC antibody and compared to MB_NT_ background signal or MB_B7-H3_ without antibody incubation. MB-ABY fluorescence signal was also analyzed using confocal microscopy (DMi8, Leica) for both targeted and non-targeted MBs (pre-labeled with CellMask Green, ThermoFisher Scientific, Waltham, MA, USA) at 20× magnification. Acquired images were analyzed using ImageJ 1.52a software (NIH, Baltimore, MD, USA).

### 3.10. In Vitro Binding Assay of MB to B7-H3

MS1_WT_ or MS1_B7-H3_ cells were cultured (3 × 10^5^ cells/well) on glass coverslips (VWR) placed on the bottom of a 6-well plate (Corning Inc, Glendale, AZ) with DMEM media (2 mL/well). After three days, non-adherent cells were removed from the wells by gently washing with warm PBS (2 mL/well; 3×) followed by application of a hydrophobic barrier (Super Pap Pen, Daido Sangyo Co., Ltd, Tokyo, Japan) around the coverslips in each well. A total of 1 × 10^8^ MBs (MB_B7-H3_ or MB_NT_) were added immediately to the top of the coverslips in warm PBS (100 µL/coverslip) to allow the MBs to bind to cells. After 30 min incubation at 37 °C, cells were gently washed with warm PBS (2 mL/well; 3×) to remove all the unbound MBs in the wells. Cells were fixed with 4% paraformaldehyde solution for 10 min and washed 3× in PBS. The dried coverslips were mounted onto glass slides with Toluene solution (Fisher Scientific, Waltham, MA, USA) and kept at 4 °C for storage. Glass slides with cell-bound MBs were imaged under brightfield microscopy (Leica DM4, 20× magnification). The number of MBs attached to MS1_WT_ or MS1_B7-H3_ cells (9 areas/coverslip) were quantified by image processing in ImageJ 1.52a. Briefly, image files were converted to 8-bit format and cell-bound MBs were identified automatically as bright circular spots by the image processing software (Spot Caliper) plugin [64] implemented via ImageJ. This experiment was repeated three times with different batches of MBs.

### 3.11. MBs Binding to B7-H3 Target Expressed in Mouse Tumor-Associated Blood Vessels after Intravenous Injection by Ex Vivo Immunostaining Analyses

Animal experiments were approved by the Institutional Administrative Panel on Laboratory Animal Care at Stanford University. MB_B7-H3_ binding to its molecular target was validated in a transgenic mouse model (FVB/N-Tg(MMTV-PyMT)634Mul/J) expressing B7-H3 in its tumor-associated vasculature [53]. A lipophilic staining reagent (CellMask Green, ThermoFisher Scientific, Waltham, MA) was used to label MBs with a fluorescent dye (Fluorescence excitation/emission maxima: 522/535 nm) prior to administration into mouse. Mammary tumors of the mouse were pre-confirmed by gene sequence testing for positive expression in all its glands (3–7 mm in diameter). Tumor positive mice (N = 10 tumors per animal) was anesthetized with a constant supply of 2% isoflurane in oxygen. A 200 μL bolus injection of MB_B7-H3_ containing 1 × 10^8^ bubbles was slowly injected intravenously through the tail vein of the mice (Catheter with 27 g butterfly needle; FUJIFILM VisualSonics, Toronto, ON, Canada) followed by 20 μL of saline flush, and MBs were allowed to attach to the molecular target, B7-H3, for 5 min. The mouse was sacrificed and infused with 5 mL of 4% paraformaldehyde (Santa Cruz Biotechnology, Dallas, TX, USA) via cardiac puncture to wash off unbound microbubbles from the circulation and to simultaneously preserve tumor tissues based on a whole animal perfusion fixation method [65]. Mouse organs and tumor tissues were surgically extracted (mammary tumor tissues, liver, kidneys) and frozen in Optimum Cutting Temperature (OCT) media (Tissue-Tek) for further processing.

The frozen tissues were sectioned (10 µm thickness) on glass slides using a Cryostat (Leica Biosystems, Wetzlar, Germany) and processed for immunofluorescence staining, as described previously [53]. Briefly, tissue sections were rinsed with PBS for 5 min to remove the OCT media. This was followed by washing for 3 times in PBS and blocking in 5% normal goat serum in PBS for one hour at room temperature. The tissue slices were further incubated with a rat anti-mouse B7-H3 primary antibody (Abcam, Waltham, MA, USA) at a dilution of 1:50 overnight at 4 °C followed by incubation with Alexa Fluor 594 goat anti-rat secondary antibody (Invitrogen, Waltham, MA, USA) at 1:300 dilution for 30 min at room temperature. The fluorescent images, representing MB localization (green channel) and B7-H3 expression (red channel), were acquired by confocal microscopy with 20× magnification (LSM 510 Meta confocal microscope, Carl Zeiss, Jena, Germany) and composite images were created in ImageJ [66].

### 3.12. Statistical Analysis

The number of MBs in the in vitro cell binding assay groups is presented as average ± standard error of mean. Groups for MB binding are compared against each other by unpaired *t*-test with a *p*-value < 0.05 considered as statistically significant.

## 4. Conclusions

In summary, we have identified a novel ABY binder protein and engineered it with cystine at C-terminus followed by a 5-glycine linker (GGGGGC) for the site-specific bioconjugation to lipid moiety for MB preparation. In addition, we added an enterokinase enzyme cleavable linker prior to His-Tag so that it can be removed after ABY protein purification. We have evaluated this ABY for its binding affinity and optimized the site-specific conjugations efficiency with DSPE-PEG to form targeting ABY-DSPE-PEG. This ABY-lipid was used to link B7-H3 for targeted MBs (MB_B7-H3_). This product was further evaluated for its binding affinity in both in vitro and in vivo studies. Finally, we have established a reproducible standardized method for generating TMBs using a microfluidic system for on-demand preparation of MBs with the potential for human studies. We performed a rigorous quality check of the engineered ABY binder, bioconjugation chemistry, and lipid formulations, while the ligand, bioconjugate, and targeted MBs were evaluated for target binding using various in vitro and in vivo studies. The in vivo efficacy and stability of these contrast agents will be examined by ultrasound molecular imaging applications in our future studies. We will further optimize and characterize these MB_B7-H3_ in preparation for US-FDA approval. In summary, MB_B7-H3_ has the potential to be used as an ultrasound contrast agent for the diagnosis of human solid tumors expressing vascular B7-H3. Our study will serve as a platform to build single- or multi-targeted ultrasound contrast agents against novel biomarkers in a highly scalable manner using microfluidic production methods, thereby enabling clinical studies.

## Figures and Tables

**Figure 1 ijms-24-09048-f001:**
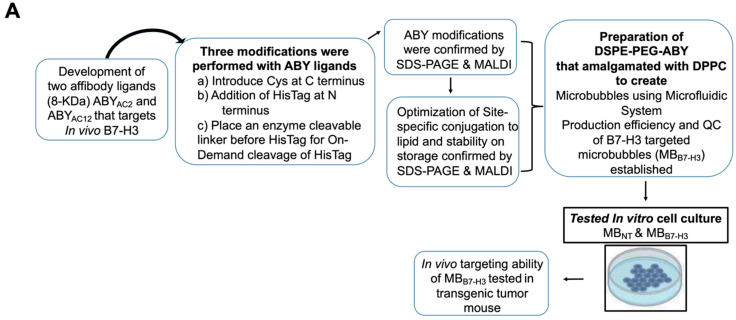
Schematic illustration of the overall study plan with the preparation of B7-H3 targeted microbubbles using a microfluidic system. (**A**) Schematic workflow of the overall study plan showing the preparation and evaluation of pharmaceutical grade B7-H3 targeted microbubbles using affibodies (ABYs). (**B**) Pictorial representation showing the preparation of targeted microbubbles using a microfluidic device.

**Figure 2 ijms-24-09048-f002:**
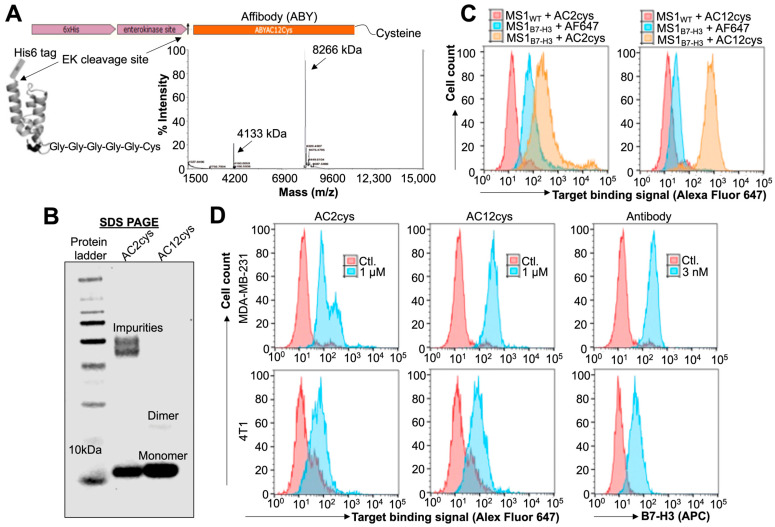
Characterization of affibodies (ABYs) with a C-terminal cysteine (cys) residue. (**A**) Schematic representation of ABY-fusion protein expression format and protein design with an N-terminal His-Tag followed by enterokinase cleavage (Asp-Asp-Asp-Asp-Lys) site, and affibody sequences. The ABY expresses a single C-terminal cysteine (Cys) following a penta-glycine (Gly-Gly-Gly-Gly-Gly) bridge. MALDI-TOF graph of AC12 ABY shows peaks representing a singly charged species (8.266 kDa) and a doubly charged species (4.1 kDa). (**B**) Comparison of two B7-H3-specific ABY_cys_ (AC2 and AC12) protein ligands resolved in of SDS-PAGE and stained by Coomassie blue. AC12 expression analysis exhibits higher purity when compared to the AC2. Ligand expression shows primarily a monomer protein band at ~8 kDa size. (**C**) Flow cytometry-based binding comparison of two ABY ligands (1 µM) to the wild-type (MS1_WT_) and B7-H3-expressing (MS1_B7-H3_) endothelial cell lines. Biotinylated-AC12 shows higher target-binding specificity compared to the biotinylated-AC2 ligand as detected by streptavidin Alexa Fluor 647 (AF647) interaction. (**D**) Flow cytometry-based binding of biotinylated-ABY ligand (1 µM) to cell-surface B7-H3 isoforms expressed by human (MDA-MB-231) and murine (4T1) breast cancer cell lines. Anti-B7-H3 APC antibody staining was used as positive control, which shows expression levels of corresponding B7-H3 isoforms in these cell lines. AC12 shows superior binding affinity to human isoform while showing the binding strength to murine isoform of B7-H3 comparable to the AC2 ligand.

**Figure 3 ijms-24-09048-f003:**
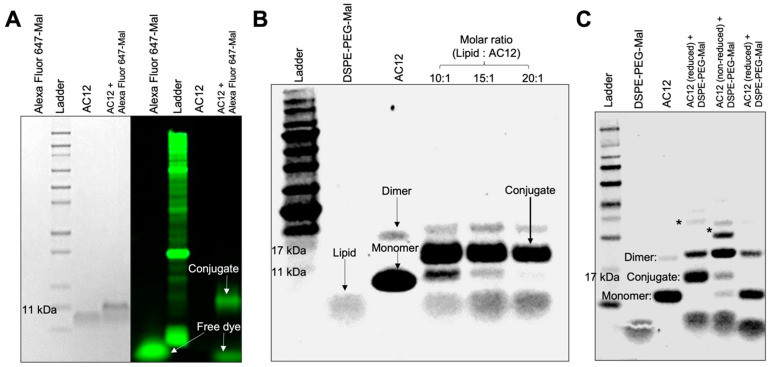
Site-specific conjugation of ABY-cys with lipid–maleimide (Mal). (**A**) Coomassie blue staining (left) and fluorescence imaging (right) of SDS-PAGE gel confirming conjugation of the Mal derivative of Alexa Fluor 647 dye (~1.2 kDa) with the reduced form (reactive free thiol group) of the AC12 ligand at 10:1 molar ratio. The unlabeled ABY band does not show fluorescence signal. (**B**) Coomassie blue staining of SDS-PAGE gel confirming conjugation of DSPE-PEG-Mal with TCEP reduced AC12 (10:1, 15:1, and 20:1 molar ratios) incubated at room temperature for 2 h. A strong band representing the conjugate (>11 kDa) formed above the AC12-cys monomer under these conditions. Staining also shows diffused bands of lipids (~2.9 kDa) and a small proportion of ABY dimer (>17 kDa). (**C**) Coomassie blue staining of SDS-PAGE gel examining thiol–maleimide reaction specificity by conjugation of DSPE-PEG-Mal with thiol reduced or non-reduced forms of AC12 (20:1 molar ratio), and a control conjugation reaction with DSPE-PEG (no Mal functional group) and reduced form of AC12, incubated at 4 °C for 16 h. This figure indicates the reduced AC12 conjugated with DSPE-PEG-Mal but not with the control DSPE-PEG, while partial conjugation was achieved with the non-reduced AC12. Undesirable products, such as ABY aggregates (dimers) and its non-specific phospholipid conjugation (*) were observed in high proportions, especially with the non-reduced ABY form, under these reaction conditions.

**Figure 4 ijms-24-09048-f004:**
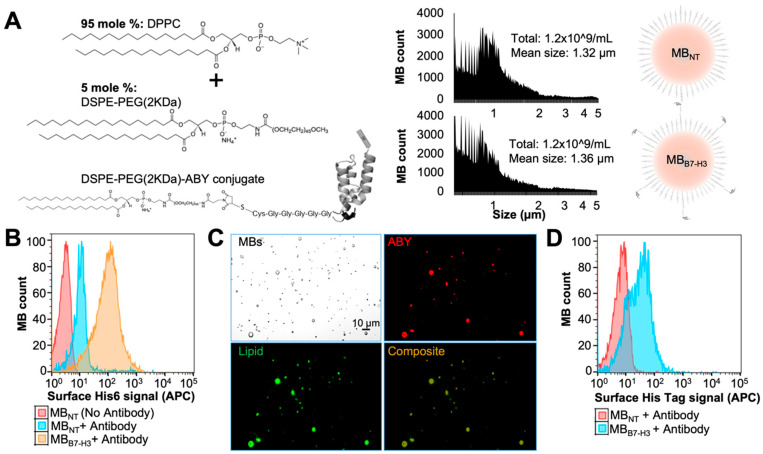
Production of targeted microbubbles (MBs) using a microfluidic system. (**A**) Left: MB formulation scheme representing phospholipid mixture, DPPC (95 mole %) and DSPE-PEG or DSPE-PEG-ABY bioconjugate (5 mole %). Right: Histograms showing mean MB size (mean diameter: ~1.3 µm) and concentration (~1.2 × 10^9^/mL) for control MBs without the bioconjugate (MB_NT_; upper graph) and targeted MBs with the ABY bioconjugate (MB_B7-H3_; lower graph), produced using a microfluidic system (Horizon Microbubble Maker, University of Leeds, U.K. (**B**) Histograms showing flow cytometry-based confirmation of His-Tag tagged ABY displayed on the targeted MB (MB_B7-H3_) shell (produced using microfluidics) by anti-His-Tag-APC antibody staining. Background staining of anti-His-Tag APC is low for the control MB_NT_. (**C**) Fluorescence microscopy-based signal confirmation of MB_B7-H3_ labeled with Cell Mask dye against phospholipid shell (green) and anti-His-Tag-APC antibody against ABY (red). The composite image shows overlap (yellow) between MBs and ABY signals. (**D**) Histograms showing the flow cytometry confirmation of His-Tag tagged ABY displayed on the MB_B7-H3_ produced by mechanical agitation method (VialMix).

**Figure 5 ijms-24-09048-f005:**
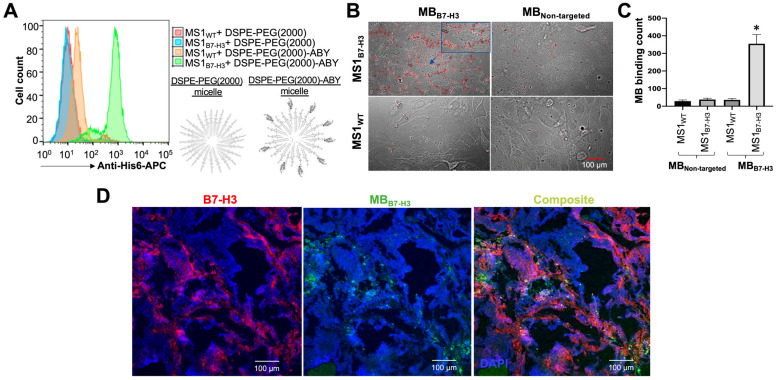
In vitro and in vivo binding validation of targeted microbubbles. (**A**) Flow cytometry assay was performed with DSPE-PEG (control) and DSPE-PEG-ABY using MS1_WT_ and MS1_B7-H3_ cells followed by anti-His-Tag-APC antibody staining. The DSPE-PEG-ABY shows strong and specific binding to the MS1_B7-H3_ cells. (**B**) Phase contrast microscopic images showing enhanced binding of MB_B7-H3_ (red circles) to a monolayer of MS1_B7-H3_ cells (upper panels) compared to the MS1_WT_ cells (lower panels). MB_NT_ binding was low for both types of cells. (**C**) Representative bar graph showing statistically significant (marked by ‘*’) counts of MB_B7-H3_ binding (red circles) to MS1_B7-H3_ cells compared to all other experimental groups. (**D**) Confocal microscopic images representing tissue immunofluorescence staining for vascular B7-H3 (red channel, Alexa Fluor 594) and intravenously injected MB_B7-H3_ (green channel) pre-labeled with fluorescent dye (Cell Mask Green), within the tumor tissues of a transgenic breast cancer mouse (MMTV-PyMT). Composite images confirm the localization of MB_B7-H3_ to tumor tissue expressing the vascular B7-H3 target. DAPI (blue) represents nuclear staining for cells.

## Data Availability

Supporting reported results can be found at Appendix A.

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
