# Peer review of "Synthesis and Evaluation of Clinically Translatable Targeted Microbubbles Using a Microfluidic Device for In Vivo Ultrasound Molecular Imaging"

_ijms, 2023, doi:10.3390/ijms24109048_

Round 1

Reviewer 1 Report

Interesting investigation. 

Figure 1 shows part of the Methods. This figure can be located in Section 3. MATERIALS AND METHODS. 

In Section 2.1, from lines 126 to 140, you describe methodology again.

The methods are well described. But the manuscript has little information about the device to generate the microbubbles and less about its operating condition. So please consider including more information.

The merit of the research lies more in the intended application. Considering the justifications for using the microfluidic device, greater emphasis should also be placed on the impact of using this type of apparatus (beyond the uniformity of the microbubbles), especially in the Results section and Conclusions. Are there studies to compare results when a device of this type is not used? This would help readers understand the advantages and disadvantages of the presented method.

Part of the Conclusions has technical language that a wider non-expert audience could not understand clearly.

There are extra spaces between periods and words, or between words. Please verify. See for example Lines 43, 47, 98, 134, 142, 158, …

 You should maintain consistency with spaces between numbers and units. See for example, 2 kDa in Line 111 and 4.13 kDa in Line 158, or 4°C in Line 245 and 4 °C in Line 260

Check parenthesis in Line 287

Author Response

We would like to thank the Reviewers for their valuable comments and suggestions. We have revised the manuscript as per suggestions, and our point-by-point responses to all the comments are given below in “Italic fonts” and addressed respectively in the revised manuscript.

  1. Figure 1 shows part of the Methods. This figure can be located in Section 3. MATERIALS AND METHODS.

We appreciate the reviewer’s suggestion. Since figure 1 is important for explaining the whole concept of the manuscript, instead of moving it to the methods section, we have kept at the beginning of the manuscript as it is currently, and additionally cited this again in the MATERIALS AND METHODS section of the revised manuscript.

  1. In Section 2.1, from lines 126 to 140, you describe methodology again.

Thank you. In the revised manuscript, we have removed some of the text to avoid any redundancy.

  1. The methods are well described. But the manuscript has little information about the device to generate the microbubbles and less about its operating condition. So please consider including more information.

We have included the following explanation of the Horizon device and its bubble generation strategy in the revised manuscript:

“The Horizon system is a microfluidics-based system for producing uniform and reproducible distributions of polydisperse and monodisperse microbubbles by mixing lipids and gases via (interchangeable) microfluidic cartridges. This computer-controlled system can produce bubble of different sizes by adjusting the flow rate of lipid mixture, gas pressure, and perfluorobutane level (see Figure 2). Similarly, the system can also be operated using cartridges of various designs to prepare MBs of different sizes and properties. A detailed design and operation of the Horizon system is reported by Abou-Saleh et al., [64, 65].”

  1. The merit of the research lies more in the intended application. Considering the justifications for using the microfluidic device, greater emphasis should also be placed on the impact of using this type of apparatus (beyond the uniformity of the microbubbles), especially in the Results section and Conclusions.

Are there studies to compare results when a device of this type is not used? This would help readers understand the advantages and disadvantages of the presented method.

We thank the reviewer for the valuable suggestion. Since this device is recently developed to produce MBs, not many studies were presented regarding the usage of this device to produce MBs for human application have been published. As per reviewer’s suggestion, we have now provided the data supporting the homogeneity of MBs produced by the Horizon microfluidic system compared to vialmix based amalgamation method (see figure below, and included this data as Figure S4 in the supplemental file).

(Figure was not able to inlude here, please see the attached file)

Legend for the above figure: MBs were prepared by two methods (vialmix and microfluidic) and compared them for bubbles count and size using acoustic spectroscopy for particle size measurement. Panel A, vial mixing preparation; panel B (Non-targeted), panel C (Targeted) produced from Horizon system. Acoustic spectroscopy clearly shown that MBs produced from Horizon system was uniform and narrow size when compared to the vial mixing preparation.

  1. Part of the Conclusions has technical language that a wider non-expert audience could not understand clearly.

As per the reviewer’s suggestion, we have modified the conclusions such that a non-expert can understand the message clearly.

  1. Page 5 of 5 Comments on the Quality of English Language There are extra spaces between periods and words, or between words. Please verify. See for example Lines 43, 47, 98, 134, 142, 158, You should maintain consistency with spaces between numbers and units. See for example, 2 kDa in Line 111 and 4.13 kDa in Line 158, or 4°C in Line 245 and 4 °C in Line 260 Check parenthesis in Line 287

Thanks, we have carefully fixed all these typos and space issues in the revised manuscript.

Reviewer 2 Report

In this manuscript, Bam et al. reported a synthesis method to prepare targeted MBs (TMBB7-H3). These TMBs showed a high binding affinity to vascular endothelial cells expressing B7-H3. However, a few experiments should be included to support their claims that these TMBs have the potential for clinical translation as a molecular ultrasound contrast agent and the necessity for microfluidic synthesis. 

1.      The authors should provide more information on the B7-H3 receptor. For example, why it is selected as a target in the current study, and if the method can be exploited for imaging other types of tumors since B7-H3 receptor is also expressed in other tumors (Int J Biol Sci. 2020 Mar 25;16(11):1767-1773; Cell Res 27, 1034–1045 (2017)).

2.      How was the “clinical grade” defined?

3.      In the abstract, the authors stated that “The synthesized MBs showed higher affinity to target in vitro and in vivo”. Please rewrite the sentence to be clear.

4.      To support their choice of microfluidic synthesis, the authors should compare the properties and efficacy (such as size distribution, targeting efficiency, etc.) of the MBs synthesized by microfluidics and traditional approach (e.g., mechanical agitation method).

5.      The authors should also show the results of ultrasound molecular imaging to support their claim that the MBs have the potential for clinical translation as a molecular ultrasound contrast agent for human applications.

6.      The authors may provide information on the MB clearance. How long do the MBs stay in the tumor site?

7.      Fig.5B and D, and Fig. S4 lack of scale bars. 

Author Response

We would like to thank the Reviewers for their valuable comments and suggestions. We have revised the manuscript as per suggestions, and our point-by-point responses to all the comments are given below in “Italic fonts” and addressed respectively in the revised manuscript

In this manuscript, Bam et al. reported a synthesis method to prepare targeted MBs (TMB). These TMBs showed a high binding affinity to vascular endothelial cells expressing B7-H3. However, a few experiments should be included to support their claims that these TMBs have the potential for clinical translation as a molecular ultrasound contrast agent and the necessity for microfluidic synthesis.

  1. The authors should provide more information on the B7-H3 receptor. For example, why it is selected as a target in the current study, and if the method can be exploited for imaging other types of tumors since B7-H3 receptor is also expressed in other tumors (Int J Biol Sci. 2020 Mar 25;16(11):1767-1773; Cell Res 27, 1034–1045 (2017)).

We thank the reviewer for this important suggestion. We have provided additional information related to B7-H3 receptor in line # 119 to 128 of the revised manuscript as shown below:

“B7-H3 (CD276) is a type I transmembrane protein of 316 amino acids (45–66 kDa) and belongs to a family of immune checkpoint molecules [24]. Several studies have shown that B7-H3  plays a major role either as a co-stimulatory or as a co-inhibitory molecule in T cell-mediated adaptive immunity [25]. However, when compared to tumor tissue, the level of expression of B7-H3 is much lower in most other normal organs and tissues. In addition to tumors, it is predominantly expressed on the surface of T and B cells [24, 26]. It has been reported that several cancer types overexpress B7-H3 [20, 27], which include, prostate cancer [28, 29], renal cell carcinoma [30], ovarian cancer [31], glioblastoma [32], osteosarcoma [33], pancreatic cancer [34], neuroblastoma [35], diffuse intrinsic pontine glioma, and mesothelioma [36, 37].”

  1. How was the “clinical grade” defined?

Non-targeted microbubbles are currently used in the clinic as contrast agents in vascular perfusion imaging applications. The main strategy needed to develop targeted contrast MBs, as we have shown here for B7-H3 target, requires the preparation of binding ligands without any associated immunogenic tags along with a clinically feasible conjugation protocol to display ligands on the MBs surface. To fulfill these requirements, in this study we have developed B7-H3 specific AC12 ligand with nanomolar affinity without any tag while having a single Cys residue at the C-terminus to specifically conjugate DSPE-phospholipid using a maleimide chemistry. Based on these reasons, we claim that AC12 based B7-H3 targeted MBs developed in this study can be translated for clinical application after addressing the toxicity studies.

We claimed our TMB as “clinical grade” since we performed several quality tests essential for human studies. However, we did not claim as “clinical use product” that requires FDA approval. We are planning to translate this TMBs for human studies seeking approval from USFDA for IND.

  1. In the abstract, the authors stated that “The synthesized MBs showed higher affinity to target in vitro and in vivo”. Please rewrite the sentence to be clear.

We appreciate the reviewer’s suggestion to revise the statement related to the in vitro and in vivo binding affinity of MBs for clarity. The synthesized MBs showed higher affinity to MS1 endothelial cells engineered to overexpress human B7-H3 in vitro in cell culture studies. Similarly, we also showed by immunofluorescence study of tumor tissues collected from animal after injection of TMBs that this MB binds to tumor specific endothelial cells in vivo. Now we have modified this sentence as printed below for clarity.

“The synthesized MBs showed higher affinity to MS1 cells expressing higher level B7-H3, and in the endothelial cells of mouse tumor tissue upon injecting TMBs in living animals.”

  1. To support their choice of microfluidic synthesis, the authors should compare the properties and efficacy (such as size distribution, targeting efficiency, etc.) of the MBs synthesized by microfluidics and traditional approach (e.g., mechanical agitation method).

As per reviewer’s suggestion, we have compared the properties of the MBs synthesized by microfluidics system with the traditional vial mix based amalgamation approach. In the revised manuscript vialmix data was included in the supplementary file as figure S4. In addition, here we have provided the comparative bubble size measurement results for the reviewers’ reference only.

(Figure was not able to inlude here, please see the attached file)

Legend for the above figure: MBs were prepared by two (vialmix and microfluidic) approaches and compared the bubbles count and size using acoustic spectroscopy for particle size measurement. A: MBs prepared by vial mix; B and C: Non-targeted and Targeted MBs prepared by Horizon microfluidic system. Acoustic spectroscopy clearly shown that MBs prepared by Horizon microfluidic system resulted with MBs of uniform and narrow size distribution compared to the vial mix based preparation.

  1. The authors should also show the results of ultrasound molecular imaging to support their claim that the MBs have the potential for clinical translation as a molecular ultrasound contrast agent for human applications.

We thank the reviewer for this important suggestion. We are currently working on extending the characterized TMBs synthesized by microfluidic system for in vivo contrast imaging in transgenic animals (FVB/N-Tg (MMTV-PyMT)634Mul/J) that spontaneously develop breast cancer. It is an extensive and time-consuming study, which requires breeding animals, screening for positive female animals, and getting enough animals with tumors for imaging applications. While waiting for all these in vivo results, we want to first publish the results related to the TMB preparations using a microfluidic system and its preliminary characterizations as this information would be useful to other researchers.

  1. The authors may provide information on the MB clearance. How long do the MBs stay in the tumor site?

We appreciate the reviewer’s suggestion. We will include these in vivo characterization results in our in vivo imaging study where we are fully evaluating the imaging sensitivity, TMBs clearance compared to non-targeted MBs, and ex vivo correlation for B7-H3 expression in comparison to imaging signal. 

  1. Fig.5B and D, and Fig. S4 lack of scale bars.

 In the revised manuscript scale bars were included.

Round 2

Reviewer 2 Report

The authors addressed my concerns well, and I do not have further questions.